# The NIH Lipo-COVID Study: A Pilot NMR Investigation of Lipoprotein Subfractions and Other Metabolites in Patients with Severe COVID-19

**DOI:** 10.3390/biomedicines9091090

**Published:** 2021-08-26

**Authors:** Rami A. Ballout, Hyesik Kong, Maureen Sampson, James D. Otvos, Andrea L. Cox, Sean Agbor-Enoh, Alan T. Remaley

**Affiliations:** 1Lipoprotein Metabolism Section, Translational Vascular Medicine Branch, National Heart, Lung and Blood Institute (NHLBI), National Institutes of Health (NIH), Bethesda, MD 20892, USA; rami-ballout@hotmail.com; 2Genomic Research Alliance for Transplantation (GRAfT) and Laboratory of Applied Precision Omics, National Heart, Lung and Blood Institute (NHLBI), National Institutes of Health (NIH), Bethesda, MD 20892, USA; hyesik.kong@nih.gov; 3Clinical Center, Department of Laboratory Medicine, National Institutes of Health (NIH), Bethesda, MD 20892, USA; msampson@cc.nih.gov; 4Laboratory Corporation of America Holdings (LabCorp), Morrisville, NC 27560, USA; otvosj@labcorp.com; 5Department of Medicine, Johns Hopkins University School of Medicine, Baltimore, MD 21205, USA; acox@jhmi.edu

**Keywords:** cholesterol, triglycerides, LDL, HDL, lipoprotein X (LpX), lipoprotein Z (LpZ), GlycA, lipids, SARS-CoV-2

## Abstract

A complex interplay exists between plasma lipoproteins and inflammation, as evidenced from studies on atherosclerosis. Alterations in plasma lipoprotein levels in the context of infectious diseases, particularly respiratory viral infections, such as SARS-CoV-2, have become of great interest in recent years, due to their potential utility as prognostic markers. Patients with severe COVID-19 have been reported to have low levels of total cholesterol, HDL-cholesterol, and LDL-cholesterol, but elevated levels of triglycerides. However, a detailed characterization of the particle counts and sizes of the different plasma lipoproteins in patients with COVID-19 has yet to be reported. In this pilot study, NMR spectroscopy was used to characterize lipoprotein particle numbers and sizes, and various metabolites, in 32 patients with severe COVID-19 admitted to the intensive care unit. Our study revealed markedly reduced HDL particle (HDL-P) numbers at presentation, especially low numbers of small HDL-P (S-HDL-P), and high counts of triglyceride-rich lipoprotein particle (TRL-P), particularly the very small and small TRL subfractions. Moreover, patients with severe COVID-19 were found to have remarkably elevated GlycA levels, and elevated levels of branched-chain amino acids and beta-hydroxybutyrate. Finally, we detected elevated levels of lipoproteins X and Z in most participants, which are distinct markers of hepatic dysfunction, and that was a novel finding.

## 1. Introduction

Despite the rapid development of several efficacious vaccines against severe acute respiratory syndrome coronavirus-2 (SARS-CoV-2) [1], the viral cause of the ongoing coronavirus disease 2019 (COVID-19) pandemic, our understanding of the potential long-term metabolic sequalae of COVID-19 remains limited [2,3,4]. A prime example is insulin resistance and thus new-onset diabetes [5,6,7,8], which have been reported to develop during infection or shortly after recovery [9]. Lipoprotein metabolism is known to be linked to insulin sensitivity, which can be reflected by changes in lipoprotein subfractions [10]. Nonetheless, to date, all available studies investigating changes in lipoprotein metabolism in COVID-19 relied only on routine diagnostic testing of plasma lipids, i.e., the standard lipid panel with or without apoA-I and apoB levels [11,12,13,14,15,16].

We conducted this pilot study at the Clinical Center of the National Institutes of Health (NIH) to better characterize the changes seen in lipoproteins and other metabolites, in individuals with severe COVID-19, using a clinically deployed NMR analyzer capable of measuring the particle numbers and sizes of different lipoprotein subfractions [17]. Moreover, it allowed us to quantify the plasma levels of several small molecule metabolites, such as branched-chain amino acids (BCAAs) and ketone bodies [18,19,20]. Finally, we also determined the levels of GlycA, a composite NMR index of overall plasma protein N-glycosylation, previously established as a marker of systemic inflammation [21,22].

## 2. Methods

Plasma samples used in this study (Johns Hopkins Medicine, Protocol # IRB00245545, 30 March 2020) were collected from 32 consenting individuals who were admitted to the intensive care units (ICU) of the Johns Hopkins University Hospital and/or the University of Maryland Medical Center, Baltimore, MD, USA, with a diagnosis of severe COVID-19 (diagnosed by a positive SARS-CoV-2 PCR result of nasal swab) that required intubation and extracorporeal membrane oxygenation (ECMO). The blood samples were collected in EDTA tubes (5 or 10 mL BD vacutainer^®^ K2 7.2 mg EDTA, Becton-Dickinson and company, Franklin Lakes, NJ, USA) at the time of patient admission and processed within 2 h of collection, followed by plasma storage in cryovials at −80 °C. Samples were thawed on ice for 3 to 4 h prior to performing their *NMR LipoProfile*^®^ analysis, using the *LP4* deconvolution algorithm on the Vantera^®^ NMR Clinical Analyzer (LabCorp^®^, Burlington, NC, USA) [23].

## 3. Results

### 3.1. Patient Characteristics

The characteristics of the participants are summarized in Table 1.

Briefly, all patients were admitted to the ICU and required ECMO for variable lengths of time. Twelve patients expired during their hospital stays due to a variety of complications, and 14 were discharged after prolonged ICU stays (median hospital stay of 42 days, with a range of 19 to 78 days). Six subjects were still in the ICU at the time of data analysis. Only 5 patients were females (~16%), which is consistent with prior reports of increased severity of COVID-19 in males [24]. Five patients were African American, twenty-three were Hispanic (i.e., Latinos), three were Caucasians, and one was African American and Hispanic. Participants’ ages ranged from 20 to 58, with a median of 42. Twenty-one patients (~66%) were obese (i.e., BMI ≥ 30), nine of whom (~28%) qualified as morbidly obese (i.e., BMI ≥ 40). Eight patients had diabetes mellitus and hypertension, two had diabetes mellitus only, three had asthma, one had chronic kidney disease (CKD), and one had cancer (Table 1). The raw data of all parameters assayed in the included patients are provided as a courtesy by the authors, in an excel sheet submitted as a Appendix A. Moreover, this manuscript was previously posted as a non-peer reviewed preprint that is publicly accessible [25]. As such, the final, published peer-reviewed version will likely differ from that preprint.

### 3.2. Significant Findings Detected in the Plasma of Participants


(a)Lipoprotein particle numbers and size distribution


A striking observation of this study was finding that more than 90% of our patients (i.e., 29 out of 32) had very low numbers of HDL particles (HDL-P), primarily due to a reduction in the number of small HDL particles (S-HDL-P; 93.8% of patients), and to a lesser degree, a reduction in the number of medium-sized HDL particles (M-HDL-P; 37.5% of patients). This finding, i.e., a reduced number of HDL particles, concurs with prior studies reporting low HDL-C levels in patients with COVID-19 [12,13].

Conversely, the number of TG-rich lipoprotein particles (TRL-P) was elevated in nearly half of our patients (14 of 32; 43.8%), primarily due to underlying increases in the numbers of small and very small subfractions (i.e., S-TRL-P and VS-TRL-P, 31.3% and 34.4%, respectively). This, too, is consistent with prior studies that showed elevated plasma TG levels in patients with COVID-19 [12,13] (Table 2).

On the other hand, most patients had total LDL particle numbers (LDL-P) falling within the reference range, despite the numbers of large and medium-sized LDL particles (L-LDL-P and M-LDL-P)—which transport most of the cholesterol carried by LDL particles—being reduced overall based on their corresponding mean and median values in our patients. Moreover, about one-third of our patients also had reduced numbers of small LDL particles (S-LDL-P; 43.8%). Most strikingly, however, nearly 80% of patients (25 of 32) had detectable levels of lipoprotein Z particles (LpZ), which are abnormal TG-enriched LDL particles that characteristically accumulate in the plasma of patients with alcoholic hepatitis [26]. In fact, LpZ particles were found to account for most LDL particles detected in our patients (median value of 1280 nmol/L; interquartile range (594 to 1907 nmol/L)) (Table 2). LpZ particles, formerly referred to as “lipoprotein B particles,” are essentially small LDL particles highly enriched in TG, phospholipids and free cholesterol, with a relatively lower overall content of esterified cholesterol and apolipoproteins, compared with normal LDL particles [27]. Normally, LpZ particles are undetectable in the plasma of healthy individuals but have been characteristically described to occur in patients with alcoholic liver disease [27]. In addition to LpZ, we also detected the presence of lipoprotein X particles (LpX) in more than 70% of participants (i.e., 23 of 32). The median value was 191 mg/dL (interquartile range 0 to 253 mg/dL). LpX is another abnormal lipoprotein particle that is usually undetectable in healthy individuals, but it can be found in patients with cholestatic liver disease or LCAT deficiency [28,29].

Since obesity is a known risk factor for altered lipoprotein metabolism, we also subdivided our data by patient BMI, into two categories: patients of healthy weight (those with BMI < 25 kg/m^2^) and obese patients (BMI ≥ 30 kg/m^2^) (Table 3).


(b)Other metabolites


At the time of admission, 24 patients were found to have elevated GlycA levels that exceeded the upper limit of the reference range (i.e., greater than 569 µmol/L). Similarly, one-half and one-quarter of the patients had elevated levels of leucine and valine (16 and 8 of 32, respectively), and nearly one-fifth had elevated levels of isoleucine (7 of 32). Almost three-quarters of patients (23 of 32; 71.9%) had very low plasma levels of alanine at the time of admission (Table 4).

While one-quarter of the included patients (8 of 32) had elevated levels of beta-hydroxybutyrate (BHB), many patients had paradoxically low levels of acetone and acetoacetate (8 and 19 of 32, respectively) (Table 4).

## 4. Discussion

In this pilot study, we investigated the changes occurring in plasma lipoprotein particle numbers and size distribution, along with several other small-molecule metabolites, in patients with severe COVID-19. We found that, at their time of admission, nearly all patients had a remarkably low levels of HDL-P, owing to a predominantly reduced count of S-HDL-P. Interestingly, S-HDL-P have been shown to exert potent anti-inflammatory, anti-thrombotic, and anti-oxidant roles [30]. Individuals with low levels of these particles appear to have a higher risk of cardiovascular disease (CVD) and CVD-associated mortality [31,32,33], an increased risk of severe inflammation [34,35], and overall shortened longevity [36]. Moreover, a study evaluating survival in patients with pulmonary arterial hypertension found that patients with low numbers of S-HDL-P had higher mortality rates; they attributed this finding to the key role of small HDL particles as carriers of fibrinolytic proteins [37]. These findings are also relevant to COVID-19, given the latter’s association with thrombophilia and thromboembolic events [38].

Likewise, an elevated number of TRL-P, as seen in our cohort, was previously reported as an independent risk factor for CVD, chronic inflammation and all-cause mortality [39,40]. Specifically, elevated numbers of TRL-P remnants, which essentially comprise the very small and small TRL-P (S-TRL-P and VS-TRL-P), have been associated with increased CVD risk, and an overall pro-inflammatory and pro-thrombotic state [40].

More than three-quarters of our patients had detectable plasma levels of LpZ particles, which are essentially TG and free cholesterol-enriched, and cholesteryl ester-depleted small LDL particles, not normally found in healthy individuals. LpZ particles have been previously detected in plasma from patients with alcoholic liver disease mainly, and patients with hypertriglyceridemia to a lesser extent [26,27]. Likewise, nearly three quarters of our patients had detectable plasma levels of LpX particles, which are essentially multilamellar vesicles of free cholesterol and phospholipids, with albumin as their core protein. LpX particles have been classically described in patients with cholestatic liver disease [28] or LCAT deficiency [29]. LpZ particle levels are thought to correlate with the degree of hepatocyte injury in patients with alcoholic liver disease, while also acting as promoters of hepatic injury, partially through their elevated contents of free cholesterol [27]. On the other hand, LpX particles are believed to be nephrotoxic due to the direct glomerular cytotoxicity of the free cholesterol content of these particles when deposited in the kidneys, coupled with their intrinsic pro-inflammatory activity of inducing IL-6 production [41]. Therefore, detecting either one, or both of these abnormal lipoprotein particles in the plasma of patients with severe COVID-19 comprises a particularly interesting and novel finding of our study, which is worth highlighting. Specifically, given that COVID-19 infection has been associated with liver enzyme abnormalities [42,43], elevated plasma levels of IL-6 [44], and increased risks for kidney injury and renal failure [45], LpX and/or LpZ particles may constitute key prognostic biomarkers in such patients.

Our finding that most patients had elevated GlycA levels upon admission was not surprising, given that the severity of COVID-19 infection and its eventual outcomes have been shown to correlate with circulating levels of inflammatory cytokines, such as IL-6 and TNF-alpha [44], and other inflammatory biomarkers, such as ESR, CRP, and procalcitonin [46,47]. However, whether measuring GlycA levels carries any additional utility for monitoring patients with COVID-19, compared to measuring the levels of the other inflammatory biomarkers alone, remains to be determined.

Our study also revealed that a large proportion of patients with severe COVID-19 had markedly elevated levels of BCAAs upon admission, which has been previously associated with insulin resistance and incident CVD events [48,49,50]. The clinical utility of elevated BCAA levels in systemic inflammatory conditions, such as sepsis or severe trauma, however, remains poorly characterized [51]. In contrast, we found nearly three-quarters of patients had diminished plasma levels of alanine, consistent with prior observations of an inverse correlation between circulating alanine levels and risk of sepsis [52].

Finally, while the levels of BHB at the time admission were remarkably elevated in one-fourth of our patients, the corresponding levels of the two other ketone bodies, i.e., acetone and acetoacetate (AA), were diminished in one-quarter and more than half of our subjects, respectively. These seemingly incongruent changes in BHB and AA levels, which yield a high BHB/AA ratio, interestingly mirror the ketone body pattern usually found in patients with diabetic ketoacidosis (DKA) or alcoholic ketoacidosis [53], where such a finding, i.e., an elevated BHB-to-AA ratio, has been correlated with worse outcomes [54,55]. Nonetheless, ketone bodies, especially BHB, have been also proposed as being potentially protective against respiratory viruses, including SARS-CoV-2, owing to their immunomodulatory and anti-inflammatory properties [56]. This, therefore, raises the question of whether the elevated plasma levels of BHB in patients with severe COVID-19 is an induced adaptive response to the infection, or simply the result of hypoperfusion and the multi-organ failure created by the infectious state.

## 5. Conclusions

To conclude, this is the first study conducted to date that used an advanced NMR analyzer to characterize the plasma lipoprotein particle numbers and sizes in patients with severe COVID-19. Our study showed that patients with severe COVID-19 had remarkably low HDL-P numbers, primarily as a result of a reduced number of small HDL particles, with a concurrently elevated number of TRL-P, especially the VS-TRL-P and S-TRL-P subfractions. Moreover, our study also found that patients with severe COVID-19 have markedly elevated levels of GlycA, BCAA, and BHB, along with a ketone body profile that resembles that seen in DKA. Finally, another potentially important finding of this study was detecting the presence of LpX and LpZ particles in plasma of patients with severe COVID-19, indicating the need for further investigating the potential value of these abnormal lipoprotein profiles as prognostic and/or risk-stratification markers in patients with COVID-19. Nonetheless, the clinical implications of this study remain currently limited, due to its small sample size, its retrospective nature, the limited outcome data available for the included patients, and the unknown potential effects of ECMO on various plasma metabolites, including lipoproteins [57]. Moreover, most patients in our cohort had obesity (BMI ≥ 30), which itself affects lipoproteins, and therefore constitutes a confounder. This coupled to the fact that the reference ranges of the different NMR parameters reported in our study were based on a large sample of individuals whose characteristics (i.e., BMI, comorbidities) are unknown. For that reason, a larger prospective study investigating the correlations of such changes with clinically relevant outcomes is still needed to inform us of the potential prognostic utility of lipoprotein subfractions and other metabolites for COVID-19.

## Figures and Tables

**Table 1 biomedicines-09-01090-t001:** Characteristics of the subjects (N = 32).

Parameter		Number of Patients Out of 32 (%)
Age (yrs)	20–30	2 (6.25)
	30–50	21 (65.6)
	≥50	9 (28.1)
Sex	Males	27 (84.4)
	Females	5 (15.6)
BMI (kg/m^2^)	<25	3 (9.4)
	25–30	8 (25)
	30–40	12 (37.5)
	≥40	9 (28.1)
Race/Ethnicity	African American	5 (15.6)
	Hispanic/Latino	23 (71.9)
	Caucasian or white	3 (9.4)
	Other *	1 (3.13)
Comorbid conditions	Diabetes Mellitus	10 (31.3)
	Hypertension ^^^	8 (25)
	Asthma	3 (9.4)
	CKD	1 (3.13)
	Cancer	1 (3.13)

***** African American and Hispanic/Latino. **^^^** All patients with hypertension had comorbid DM.

**Table 2 biomedicines-09-01090-t002:** Tabular summary of the lipoprotein profiles of all participants in the study.

Lipid Parameters	Number *(%)* of Subjects Outside Reference Range	Mean	Median	Interquartile Range (25th–75th Percentile)	Reference Range
**HDL-P**	29 (90.6) ↓	7 µmol/L	6.4 µmol/L	4.7–9.3 µmol/L	15.2–27.5 µmol/L
**L-HDL-P**	2 (6.3) ↓	1	1.3	0.88–1.8	0.1–6.9
**M-HDL-P**	12 (37.5) ↓	3	2.7	0.91–4.4	1.6–8.1
**S-HDL-P**	30 (93.8) ↓	3	1.6	0.18–4.6	8.2–20.6
**TRL-P**	14 (43.8) ↑	275 nmol/L	277 nmol/L	155–349 nmol/L	19–291 nmol/L
**VL-TRL-P**	1 (3.1) ↑	0	0.03	0.001–0.15	0–0.7
**L-TRL-P**	5 (15.6) ↑	8	3.64	0.063–8.06	0–11.9
**M-TRL-P**	12 (37.5) ↓	25	11	0–31	0.4–70.7
**S-TRL-P**	10 (31.3) ↑	79	69	25–116	0–103.6
**VS-TRL-P**	11 (34.4) ↑	164	154	65–239	0–184.2
**LDL-P**	4 (12.5) ↑	1890 nmol/L	1800 nmol/L	1397–2305 nmol/L	851–2585 nmol/L
**L-LDL-P**	0	63	7.1	0–90	0–674
**M-LDL-P**	0	61	0	0–35	0–1376
**S-LDL-P**	14 (43.8) ↓	549	108	0–811	71–1865
**LpZ particles**	25 (78.1) ↑	1217 nmol/L	1280 nmol/L	594–1907 nmol/L	0 nmol/L
**LpX particles**	23 (72) ↑	149 mg/dL	191 mg/dL	0–253 mg/dL	0 mg/dL

**Table 3 biomedicines-09-01090-t003:** The distribution of HDL, TRL, LpX and LpZ particles by BMI of the patients.

Body Mass Index (Kg/m^2^)	Lipid Parameters	Mean	Median	Interquartile Range (25th–75th Percentile)	Reference Range
<**25** (N = 3)	**HDL-P**	13 µmol/L	15.5 µmol/L	10.4–16.4 µmol/L	15.2–27.5 µmol/L
	**L-HDL-P**	1	1.6	1.2–1.8	0.1–6.9
	**M-HDL-P**	6	3.7	2.6–8.7	1.6–8.1
	**S-HDL-P**	5	3	2.2–6.6	8.2–20.6
	**TRL-P**	180 nmol/L	157 nmol/L	107–242 nmol/L	19–291 nmol/L
	**VL-TRL-P**	0	0.1	0.1–0.4	0–0.7
	**L-TRL-P**	3	3.9	2–5.1	0–11.9
	**M-TRL-P**	18	21	10–27	0.4–70.7
	**S-TRL-P**	45	63	34–65	0–103.6
	**VS-TRL-P**	113	55	53–144	0–184.2
	**LpZ particles**	394 nmol/L	0 nmol/L	0–592 nmol/L	0 nmol/L
	**LpX particles**	0 mg/dL	0 mg/dL	0 mg/dL	0 mg/dL
≥**30** (N = 21)	**HDL-P**	7 µmol/L	6.92 µmol/L	4.3–9.9 µmol/L	15.2–27.5 µmol/L
	**L-HDL-P**	2	1.4	0.99–1.9	0.1–6.9
	**M-HDL-P**	2	1.61	0.62–4.1	1.6–8.1
	**S-HDL-P**	3	2.73	0.06–6.5	8.2–20.6
	**TRL-P**	278 nmol/L	270.5 nmol/L	166–346 nmol/L	19–291 nmol/L
	**VL-TRL-P**	0	0.058	0.002–0.152	0–0.7
	**L-TRL-P**	10	3.43	0.26–10.4	0–11.9
	**M-TRL-P**	26	4.78	0–27.5	0.4–70.7
	**S-TRL-P**	80	76.3	25.4–116	0–103.6
	**VS-TRL-P**	162	151	67–228	0–184.2
	**LpZ particles**	1268 nmol/L	1378 nmol/L	615–1991 nmol/L	0 nmol/L
	**LpX particles**	131 mg/dL	172 mg/dL	2.89–247 mg/dL	0 mg/dL

**Table 4 biomedicines-09-01090-t004:** Tabular summary of the levels of all other metabolites assayed in our study participants.

Other Metabolites	Number *(%)* of Subjects Outside Reference Range	Mean (in µmol/L)	Median (in µmol/L)	Interquartile Range (25th–75th Percentile) (in µmol/L)	Reference Range (in µmol/L)
**Acetone**	8 (25) ↓	68	33	13–50	11–127
**Acetoacetate**	19 (59.4) ↓	19	18	12–28	21–130
**BHB**	8 (25) ↑	447	235	132–400	40–396
**Alanine**	23 (71.9) ↓	263	257	207–319	293–614
**Leucine**	16 (50) ↑	246	243	220–285	122–245
**Isoleucine**	7 (21.9) ↑	61	57	39–72	22–80
**Valine**	8 (25) ↑	282	271	219–332	173–340
**GlycA**	24 (75) ↑	710	713	571–819	312–569

## Data Availability

The algorithm used to deconvolve the NMR spectra generated for the patients included in this study remains the proprietary property of LabCorp^®^. However, data about the included participants may be provided by the corresponding authors upon reasonable request, provided that sharing such data does not jeopardize patient confidentiality.

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
