# Peer review of "The NIH Lipo-COVID Study: A Pilot NMR Investigation of Lipoprotein Subfractions and Other Metabolites in Patients with Severe COVID-19"

_biomedicines, 2021, doi:10.3390/biomedicines9091090_

Round 1
Reviewer 1 Report
In this work authors investigated lipoprotein particle numbers in patients s with severe COVID-19.
The topic is interesting, but some points should be clarified even if it is a pilot study:
- Most of patients have very high BMI. Obesity and metabolic syndrome have a significant impact on lipoprotein distribution. A correction for sex and BMI is necessary to understand the differences in lipoprotein distribution, especially for HDL. Moreover, in table I it will be helpful have not only the reference value for the general population but also the value of obese patients to have a complete picture and to discriminate the effect of COVID-19 from this confounding factor.
- In table I it was reported “BMI<30”, please discriminate the patients with BMI<25 and patients with BMI between 25 and 30.
- LpX values were not reported in table I.
Author Response
Please see attached our Cover Letter that addresses, point-by-point, your valuable suggestions and comments.

Reviewer 2 Report
This is a very interesting and an important study to understand viral infection and characterize the lipoprotein for COVID-19 patients. the following information should be provided before publication:
1)please provide some details of the virus (e.g. species) if it is possible in the patients.
2) I understand the data has been well characterized, it would be useful for other researchers if the authors should some original NMR data from the patient as supplementary data. this is a very novel study, the raw data will be helpful for other researchers.
3) Please elaborate the method part.
Author Response
Please see attached our Cover Letter that addresses, point-by-point, all the valuable comments and suggestions you kindly raised.
